# Computing mathematical functions with chemical reactions via stochastic logic

**Arnav Solanki**, **Tonglin Chen**, **Marc Riedel**\*

Department of Electrical and Computer Engineering, University of Minnesota Twin-Cities, Minneapolis, MN, United States of America

\* mriedel@umn.edu

## Abstract

This paper presents a novel strategy for computing mathematical functions with molecular reactions, based on theory from the realm of digital design. It demonstrates how to design chemical reaction networks based on truth tables that specify analog functions, computed by stochastic logic. The theory of stochastic logic entails the use of random streams of zeros and ones to represent probabilistic values. A link is made between the representation of random variables with stochastic logic on the one hand, and the representation of variables in molecular systems as the concentration of molecular species, on the other. Research in stochastic logic has demonstrated that many mathematical functions of interest can be computed with simple circuits built with logic gates. This paper presents a general and efficient methodology for translating mathematical functions computed by stochastic logic circuits into chemical reaction networks. Simulations show that the computation performed by the reaction networks is accurate and robust to variations in the reaction rates, within a log-order constraint. Reaction networks are given that compute functions for applications such as image and signal processing, as well as machine learning: arctan, exponential, *Bessel*, and sinc. An implementation is proposed with a specific experimental chassis: DNA strand displacement with units called DNA "concatemers".

**Data Availability Statement:** All relevant data are within the paper and its Supporting information files.

**Funding:** MR DARPA Grant W911NF-18-2-0032 https://www.darpa.mil The funders had no role in

## 1 Introduction

In recent years, the topic of stochastic logic has been advertised as a possible design paradigm for emerging technologies that promise scaling beyond complementary metal–oxide–semiconductor (CMOS), as well as the basis of non-von Neumann architectures [1, 2]. While the term can mean many things, ranging from randomized algorithms to probabilistic analysis, in our context "stochastic computing" or "stochastic logic" has a specific meaning: it refers to logic-level computation on randomized bitstreams. Instead of the traditional values of 1 and 0 that form the basis of binary computing systems, in stochastic computing a real value $x$ is represented as a stream of random bits. In this stream, the probability of a randomly chosen bit being 1 is $x$, and the probability of it being 0 is $1 - x$.

The original ideas for this form of stochastic computation are generally attributed to research by Gaines and Poppelbaum in the late 1960s [3, 4], as well as to work by Brown and

study design, data collection and analysis, decision to publish, or preparation of the manuscript.

**Competing interests:** The authors have declared that no competing interests exist.

Card in the 1990s [5]. Beginning in the late 2000s, there has been a renewed interest, with too many publications to enumerate. We point to some influential papers as well as surveys: [6–11]. In [12, 13], Qian et al. presented a general synthesis methodology for stochastic logic. Our exposition is based on that framework.

The main appeal of stochastic logic is that a wide variety of functions can be computed with simple structures. For instance, multiplication can be implemented with a single AND gate. More complicated functions such as the exponential, absolute value, square roots, and hyperbolic tangent can each be computed with a very small number of gates [14]. Simplicity is a compelling advantage for the task that we confront in this paper: computing with molecular reactions.

The idea of molecular computing dates back to seminal work by Len Adleman, who discussed solutions to combinatorial problems such as Boolean satisfiability and Hamiltonian paths with DNA [15]. There has been a broad range of research since. We point to a small subset: [16–22].

This paper explores a link between the two fields. Specifically, it presents a strategy for computing mathematical functions with molecular reactions by applying concepts from stochastic logic. We preview with an example. Suppose we want a chemical reaction network that computes the function

$$f(a, b) = 1 - a - b + ab,$$

where $a$ and $b$ are real-valued variables. The corresponding digital function for stochastic logic can be obtained using the methods discussed in Section 1. In this case, it is $f(a, b) = \text{NOR}(a, b)$, expressed in the following truth table:

| $a$ | $b$ | $\text{NOR}(a, b)$ |
|:---:|:---:|:---:|
| 0 | 0 | 1 |
| 0 | 1 | 0 |
| 1 | 0 | 0 |
| 1 | 1 | 0 |

To represent a stochastic variable $x$ that ranges from [0, 1] in a molecular format, we use a *pair* of chemical species $X_0$ and $X_1$. As will be discussed in Section 2, we use a *fractional* representation:

$$x = \frac{[\mathbf{X}_1]}{[\mathbf{X}_0] + [\mathbf{X}_1]}. \tag{1}$$

Here $[X_1]$ denotes the concentration of the molecular species $X_1$. Using this representation, we obtain a chemical reaction network (CRN) from the truth table above:

$$
\begin{aligned}
\mathbf{A}_0 + \mathbf{B}_0 &\rightarrow \mathbf{C}_1 \\
\mathbf{A}_0 + \mathbf{B}_1 &\rightarrow \mathbf{C}_0 \\
\mathbf{A}_1 + \mathbf{B}_0 &\rightarrow \mathbf{C}_0 \\
\mathbf{A}_1 + \mathbf{B}_1 &\rightarrow \mathbf{C}_0
\end{aligned}
\tag{2}
$$

Note that the subscripts of the species match the entries of the truth table above. This CRN computes the target function, $c = 1 - a - b + ab$, in terms of the fractional variables $a$, $b$ and $c$. Each of these corresponds to a pair of chemical species, $\{A_0, A_1\}$, $\{B_0, B_1\}$ and $\{C_0, C_1\}$, respectively. The central result of this paper, presented in Section 4, is a proof that we can implement any polynomial function, specified by a truth table, with a CRN matching its truth table template.

This paper builds upon our prior work, both generalizing and simplifying it. We use the same formalism, namely a fractional representation of values, in this paper as in [23] and [24].

- In [23], we proposed a technique for computing functions based on a decomposition with *Bernstein polynomials* [25]. The technique can implement a broad class of functions, namely all univariate polynomials, but is quite abstruse. A target polynomial is first repackaged in Bernstein form [26]. This form is implemented in a logic circuit using a form of generalized *multiplexing* [13]. Finally, the logic circuit is translated into a CRN.

- In [24], we proposed an alternative technique based on factoring of polynomials with *Horner's rule*. The factored form is implemented with a cascade of 2-input logic gates. Finally, the logic gate circuit is translated into a CRN. Although conceptually simpler than working with Bernstein polynomials, this approach is not quite so general: only a small subset of polynomials can be decomposed in the requisite way with Horner's rule.

A significant limitation of both prior approaches is the complexity of the mathematical formulation.

The approach in this paper is conceptually much simpler and cleaner. As with the NOR function example above, a target polynomial function is first mapped to a truth table. This can be done using fairly standard techniques—at least for people familiar with the theory of stochastic logic—and the results are intuitive. Then a CRN is constructed that matches the template of the truth table.

This approach is also more general. Whereas the method in [23] is limited to univariate polynomials, the method in this paper can implement any multivariate polynomial. Stochastic logic operates on functions where the domain and codomain are in the interval [0, 1], i.e., the inputs and the output are probabilities. Common transcendental functions can be computed via polynomial approximations. In S1 File, we provide CRNs for stochastic functions such as arctan, exponential, Bessel, and sinc to demonstrate our approach in detail. These functions have practical applications in fields such as machine learning, signal processing, and image processing. We discuss the implementation of these abstract chemical reaction networks with DNA strand displacement, with units called DNA concatemers.

This paper is organized as follows. Section 2 presents background information on chemical reaction networks and stochastic logic. Section 3 describes our methodology for translating any function computed by a stochastic logic circuit into a set of chemical reactions. Section 4 provides a proof that the proposed methodology is mathematically sound, based on an analysis of the chemical kinetics. Section 5 analyzes sources of error stemming from differences in reaction rates in one particular case. Section 6 discusses the implementation with DNA strand displacement. It explains how stochastic values can be mapped to DNA molecules that are capable of self-polymerizing—what we call DNA "concatemers". These concatemers implement the generic chemical reaction networks presented in the early sections. Finally, Section 7 provides concluding remarks and discusses future research directions.

## 2 Background

### 2.1 Chemical reaction networks

A chemical reaction network (CRN) consists of a set of *reactions* operating on a set of *molecular species*. When a reaction fires, *reactant* molecules are transformed into *product* molecules. For instance, consider the reaction:

$$\mathbf{X}_1 + \mathbf{X}_2 \xrightarrow{\text{k}} \mathbf{X}_3.$$

Here one molecule of reactant $\mathbf{X}_1$ combines with one molecule of reactant $\mathbf{X}_2$, resulting in one molecule of the product $\mathbf{X}_3$. The parameter $k$ is called the *rate constant*. A CRN consists of multiple reactions occurring simultaneously. Consider a toy example of a CRN with three reactions operating on the molecule species set $\{\mathbf{X}_1, \mathbf{X}_2, \mathbf{X}_3, \mathbf{X}_4\}$:

$$\mathbf{X}_1 + \mathbf{X}_2 \rightarrow \quad \mathbf{X}_3,$$

$$\mathbf{X}_2 + \mathbf{X}_3 \rightarrow \quad 2\mathbf{X}_4,$$

$$3\mathbf{X}_3 + \mathbf{X}_4 \rightarrow \quad \mathbf{X}_1.$$

Here we assume that all three reactions have the same rate constant, $k$, an arbitrary value. To quantify the changes in concentration of all the molecular species involved in a CRN over time we can apply the theory of *mass-action kinetics* [27]: reaction rates are proportional to both the concentrations of the reactants and their rate constants. Given a CRN, one can derive a set of nonlinear differential equations for the concentrations of all molecular species. For instance, for the first reaction above, the rate of change of the concentrations of $\mathbf{X}_1$, $\mathbf{X}_2$ and $\mathbf{X}_3$ is

$$-\frac{d[\mathbf{X}_1]}{dt} = -\frac{d[\mathbf{X}_2]}{dt} = \frac{d[\mathbf{X}_3]}{dt} = k[\mathbf{X}_1][\mathbf{X}_2], \tag{3}$$

where $[\mathbf{X}]$ denotes the concentration of the chemical species $\mathbf{X}$. (We omit the equations for the second and third reactions for brevity.) Given the initial concentration of the different molecular species, one can predict the behavior of the CRN by simulating the differential equations.

### 2.3 Digital logic

We give some basic definitions that we will need pertaining to digital logic.

**Definition 1 (Combinational Logic Function)** An $n$-input combinational logic function is a function $F(X_1, X_2, \ldots, X_n) = Y$, where all inputs and outputs are Boolean values. That is, $\forall 1 \leq i \leq n, X_i \in \{0, 1\}, Y \in \{0, 1\}$.

**Definition 2 (Truth table)** The truth table of a combinational logic function lists all the possible combinations of its inputs and the corresponding outputs. Each combination of inputs is called a **minterm**.

Table 1 in the next section gives an example of the truth table of a combinational logic function. (We also provided the truth table for the NOR function in Section 1.)

### 2.3 Stochastic logic

Stochastic logic is an active topic of research in digital design, with applications to emerging technologies [3, 13, 28]. Computation is performed with familiar digital constructs, such as AND, OR, and NOT gates. However, instead of having specific Boolean values of 0 and 1, the inputs are random bitstreams. A number $x$ $(0 \leq x \leq 1)$ corresponds to a sequence of random

**Table 1. Truth table for a combinational circuit, and the corresponding probability of each row.**

| $X_1$ | $X_2$ | $X_3$ | $F(X_1, X_2, X_3)$ | Probability of row |
|-------|-------|-------|---------------------|--------------------|
| 0 | 0 | 0 | 0 | $(1 - x_1) \cdot (1 - x_2) \cdot (1 - x_3)$ |
| 0 | 0 | 1 | 1 | $(1 - x_1) \cdot (1 - x_2) \cdot x_3$ |
| 0 | 1 | 0 | 0 | $(1 - x_1) \cdot x_2 \cdot (1 - x_3)$ |
| 0 | 1 | 1 | 1 | $(1 - x_1) \cdot x_2 \cdot x_3$ |
| 1 | 0 | 0 | 0 | $x_1 \cdot (1 - x_2) \cdot (1 - x_3)$ |
| 1 | 0 | 1 | 1 | $x_1 \cdot (1 - x_2) \cdot x_3$ |
| 1 | 1 | 0 | 1 | $x_1 \cdot x_2 \cdot (1 - x_3)$ |
| 1 | 1 | 1 | 1 | $x_1 \cdot x_2 \cdot x_3$ |

bits. Each bit has *probability x* of being one and probability $1 - x$ of being zero, as illustrated in Fig 1. Computation is recast in terms of the probabilities observed in these streams.

Consider basic logic gates. Given a stochastic input $x$, a NOT gate implements the function

$$\text{NOT}(x) = 1 - x. \tag{4}$$

This means that while an individual input of 1 results in an output of 0 for the NOT gate (and vice versa), statistically, for a random bitstream that encodes the stochastic value $x$, the NOT gate output is a new bitstream that encodes $1 - x$.

The output of an AND gate is 1 only if all the inputs are simultaneously 1. The probability of the output being 1 is thus the probability of all the inputs being 1. Therefore, an AND gate implements the stochastic function:

$$\text{AND}(x, y) = xy, \tag{5}$$

that is to say, multiplication. The output of an OR gate is 0 only if all the inputs are 0. Therefore, an OR gate implements the stochastic function:

$$\text{OR}(x, y) = 1 - (1 - x)(1 - y) = x + y - xy. \tag{6}$$

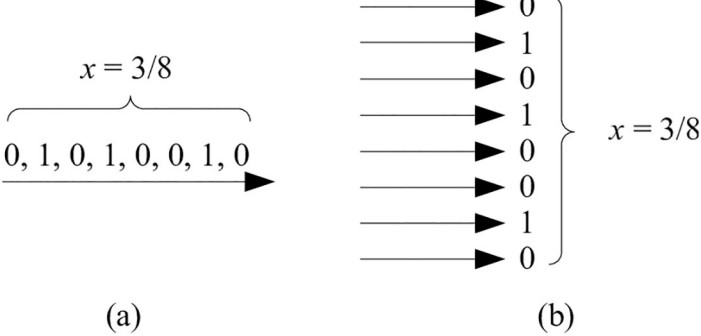

(a) (b)

**Fig 1. Stochastic representation: A random bitstream.** A value $x \in [0, 1]$, in this case 3/8, is represented as a bitstream. The probability that a randomly sampled bit in the stream is one is $x = 3/8$; the probability that it is zero is $1 - x = 5/8$.

The output of an XOR gate is 1 only if the two inputs $x$, $y$ are different. Therefore, an XOR gate implements the stochastic function:

$$\text{XOR}(x, y) = (1 - x)y + x(1 - y) = x + y - 2xy. \tag{7}$$

The NAND, NOR, and XNOR gates can be derived by composing the AND, OR, and XOR gates each with a NOT gate, respectively. Please refer to Table 2 for a full list of the algebraic expressions of these gates. An important assumption in stochastic computation is that all inputs are independent of each other, i.e., the random bitstreams are uncorrelated.

We formalize the definition of stochastic logic functions as follows.

**Definition 3 (Stochastic Logic Function)** An $n$-input stochastic logic function $y = f(x_1, x_2, \ldots, x_n)$, where $\forall x_i \in [0, 1]$ and $y \in [0, 1]$, is obtained from a combinational logic function $Y = F(X_1, X_2, \ldots, X_n)$, by setting corresponding inputs to be independent random variables $X_i$ with $\Pr(X_i = 1) = x_i$.

For a given Boolean circuit, its stochastic function can be computed as follows.

**Table 2. Chemical reaction networks for basic logic gates.** Note that the indices of molecules match the truth table implementing the logic gate.

| gate | inputs | function | CRN |
|------|--------|----------|-----|
| NOT | $a$ | $b = 1 - a$ | $\mathbf{A}_0 \rightarrow \mathbf{B}_1$ |
|  |  |  | $\mathbf{A}_1 \rightarrow \mathbf{B}_0$ |
| AND | $a, b$ | $c = ab$ | $\mathbf{A}_0 + \mathbf{B}_0 \rightarrow \mathbf{C}_0$ |
|  |  |  | $\mathbf{A}_0 + \mathbf{B}_1 \rightarrow \mathbf{C}_0$ |
|  |  |  | $\mathbf{A}_1 + \mathbf{B}_0 \rightarrow \mathbf{C}_0$ |
|  |  |  | $\mathbf{A}_1 + \mathbf{B}_1 \rightarrow \mathbf{C}_1$ |
| OR | $a, b$ | $c = a + b - ab$ | $\mathbf{A}_0 + \mathbf{B}_0 \rightarrow \mathbf{C}_0$ |
|  |  |  | $\mathbf{A}_0 + \mathbf{B}_1 \rightarrow \mathbf{C}_1$ |
|  |  |  | $\mathbf{A}_1 + \mathbf{B}_0 \rightarrow \mathbf{C}_1$ |
|  |  |  | $\mathbf{A}_1 + \mathbf{B}_1 \rightarrow \mathbf{C}_1$ |
| NAND | $a, b$ | $c = 1 - ab$ | $\mathbf{A}_0 + \mathbf{B}_0 \rightarrow \mathbf{C}_1$ |
|  |  |  | $\mathbf{A}_0 + \mathbf{B}_1 \rightarrow \mathbf{C}_1$ |
|  |  |  | $\mathbf{A}_1 + \mathbf{B}_0 \rightarrow \mathbf{C}_1$ |
|  |  |  | $\mathbf{A}_1 + \mathbf{B}_1 \rightarrow \mathbf{C}_0$ |
| NOR | $a, b$ | $c = 1 - a - b + ab$ | $\mathbf{A}_0 + \mathbf{B}_0 \rightarrow \mathbf{C}_1$ |
|  |  |  | $\mathbf{A}_0 + \mathbf{B}_1 \rightarrow \mathbf{C}_0$ |
|  |  |  | $\mathbf{A}_1 + \mathbf{B}_0 \rightarrow \mathbf{C}_0$ |
|  |  |  | $\mathbf{A}_1 + \mathbf{B}_1 \rightarrow \mathbf{C}_0$ |
| XOR | $a, b$ | $c = a + b - 2ab$ | $\mathbf{A}_0 + \mathbf{B}_0 \rightarrow \mathbf{C}_0$ |
|  |  |  | $\mathbf{A}_0 + \mathbf{B}_1 \rightarrow \mathbf{C}_1$ |
|  |  |  | $\mathbf{A}_1 + \mathbf{B}_0 \rightarrow \mathbf{C}_1$ |
|  |  |  | $\mathbf{A}_1 + \mathbf{B}_1 \rightarrow \mathbf{C}_0$ |
| XNOR | $a, b$ | $c = 1 - a - b + 2ab$ | $\mathbf{A}_0 + \mathbf{B}_0 \rightarrow \mathbf{C}_1$ |
|  |  |  | $\mathbf{A}_0 + \mathbf{B}_1 \rightarrow \mathbf{C}_0$ |
|  |  |  | $\mathbf{A}_1 + \mathbf{B}_0 \rightarrow \mathbf{C}_0$ |
|  |  |  | $\mathbf{A}_1 + \mathbf{B}_1 \rightarrow \mathbf{C}_1$ |

**Theorem 1 (Output of a Stochastic Logic Function** [6]) *Given input sequences generated by independent Bernoulli random variables, the output of a stochastic logic function will also be a sequence generated by a Bernoulli random variable. The probability of the output of a stochastic logic function f being 1 is the sum of all the probabilities of the minterms that evaluate to 1 in the corresponding combination logic function F. That is,*

$$\Pr(Y = 1) = \sum_{J \in S} \left( \prod_{h=1}^{n} [\Pr(X_h = j_h)] \right) \tag{8}$$

*where $J = (j_1, j_2, \ldots, j_n)$, $j_i \in \{0, 1\}$ is a minterm, and $S = \{J | F(J) = 1\}$ is the set of minterms that evaluate to 1.*

To elucidate Theorem 1, we step through the implementation of a stochastic logic function from a truth table. Consider a combinational circuit computing a function $F(X_1, X_2, X_3)$ with the truth table shown in Table 1. Let $f(x_1, x_2, x_3)$ be the stochastic function computed by this circuit, with real-valued inputs $x_1, x_2, x_3 \in [0, 1]$. Assuming each input is independent of the others, set

$$[Pr(X_1) = 1] = x_1, \tag{9}$$

$$[Pr(X_2) = 1] = x_2, \tag{10}$$

$$[Pr(X_3) = 1] = x_3. \tag{11}$$

The probability that the function $f$ evaluates to 1 is equal to the sum of the probabilities of occurrence of each row that evaluates to 1. The probability of occurrence of each row, in turn, is obtained from the assignments to the variables, as shown in Table 1: $x_i$ if the corresponding variable $X_i$ is 1 and $(1 - x_i)$ if it is 0. Thus, we filter the rows in Table 1 where $F(X_1, X_2, X_3) = 1$ and add their probabilities together to obtain the expression for the stochastic function:

$$\begin{aligned} f(x_1, x_2, x_3) = \quad & (1 - x_1)(1 - x_2)x_3 \quad + \\ & (1 - x_1)x_2x_3 \quad + \\ & x_1(1 - x_2)x_3 \quad + \\ & x_1x_2(1 - x_3) \quad + \\ & x_1x_2x_3 \\ = \quad & (1 - x_2)x_3 + x_2x_3 + x_1x_2(1 - x_3). \end{aligned} \tag{12}$$

The procedure shown for this example can be generalized to any combinational circuit to evaluate its stochastic function. Such probabilistic analysis of networks of logic gates is not new. As early as 1975, the circuit testing community had begun analyzing errors in a similar way [29, 30]. Similar techniques have also been applied to tasks such as timing and power analysis [31, 32]. However, characterizing the *outputs* of the computation this way, as probabilistic functions, is specific to the field of stochastic logic. We point to some of our prior work in this field. In [26] we proved that any multivariate polynomial function with its domain and codomain in the unit interval [0, 1] can be implemented using stochastic logic. In [13], we provide an efficient and general synthesis procedure for stochastic logic, the first in the field. In [8], we provided a method for transforming probabilities values with digital logic. Finally, in [11, 33] we demonstrated how stochastic computation can be performed deterministically.

## 3 Implementing stochastic logic with chemical reactions

In the introduction, we gave a brief example of translating a simple polynomial function, the NOR function, into a CRN. In this section, we step through the details of this process.

### 3.1 Fractional representation in solution

To represent a stochastic value $x$ in a chemical system, we use two distinct molecular species $\mathbf{X_0}$ and $\mathbf{X_1}$ such that

$$x = \frac{[\mathbf{X_1}]}{[\mathbf{X_0}] + [\mathbf{X_1}]}. \tag{13}$$

Here we use the notation $[\mathbf{X}]$ to refer to the concentration of a molecular species $\mathbf{X}$. We introduced this *fractional* representation in our prior work [23, 24]: the value $x$ equals the ratio of the concentration of $\mathbf{X_1}$ to the total concentration of $\mathbf{X_0}$ and $\mathbf{X_1}$. As with probabilities in stochastic logic, such a fractional value can represent any real number in the unit interval [0, 1]. Indeed, we will demonstrate how this fractional encoding can be used to compute stochastic functions. We present a potential experimental implementation using DNA strand displacement in Section 6.

### 3.2 Building a chemical reaction network from a truth table

Consider the truth table for the Boolean AND operation:

| *A* | *B* | *C* = AND(*A*, *B*) |
|---|---|---|
| 0 | 0 | 0 |
| 0 | 1 | 0 |
| 1 | 0 | 0 |
| 1 | 1 | 1 |

Given the fractional representation described above, let us design a CRN that performs multiplication with an AND operation on two stochastic inputs $a$ and $b$, producing an output $c$. The network consists of the following reactions:

$$\mathbf{A_0} + \mathbf{B_0} \xrightarrow{k} \mathbf{C_0},$$

$$\mathbf{A_0} + \mathbf{B_1} \xrightarrow{k} \mathbf{C_0},$$

$$\mathbf{A_1} + \mathbf{B_0} \xrightarrow{k} \mathbf{C_0}, \tag{14}$$

$$\mathbf{A_1} + \mathbf{B_1} \xrightarrow{k} \mathbf{C_1}.$$

Here $k$ is the rate constant, an arbitrary value, equal for all the reactions. Notice that there is a one-to-one mapping from the Boolean truth table of the AND gate to the indices of the

chemical species. Note that, given the two inputs $a$ and $b$ in the fractional encoding,

$$a = \frac{[A_1]}{[A_0] + [A_1]} \quad \text{and} \quad b = \frac{[B_1]}{[B_0] + [B_1]}. \tag{15}$$

If we simulate this CRN, we observe that

$$c = \frac{[C_1]}{[C_0] + [C_1]} = a \times b. \tag{16}$$

That is, the output value is the product of the two input values.

This strategy for implementing stochastic functions with CRN works for an arbitrary number of inputs, provided the reaction rates are the same for all reactions. We will prove this assertion in Section 4. Table 2 lists CRNs that implement the stochastic functions of all the basic logic gates. Again, note that the indices that appear in each CRN match the truth table of the corresponding gate.

The rate constants for all reactions in these CRNs must be equal for the computation to proceed correctly. Consider a different situation: for the CRN presented in Eq 14, suppose that the rate constant of the fourth reaction is $2k$, while all the other rate constants are $k$ (where $k$ is an arbitrary value). Given stochastic inputs $a = 0.7$ and $b = 0.6$, simulation shows that the output is $c = 0.462$ instead of the expected value $a \times b = 0.42$. We analyze the effects of varying rate constants on the accuracy of the computation in Section 5.

We note that the number of reactions in a CRN that we design equals the number of rows in the truth table of the corresponding function. The number of rows in a truth table is, of course, exponential in the number of variables: with $n$ variables there are $2^n$ rows. So, in principle, the approach that we suggest here could lead to CRNs with an unmanageable number of reactions. However, as was noted in Section 1, stochastic logic permits a wide range of complex functions to be implemented with very simple logic [13, 14]. In S1 File, we provide CRNs for computing polynomial approximations for functions such as arctan, exponential, Bessel, and sinc. All of these are computed by truth tables with 4, 5 or 6 variables. In the field of molecular computing, there is essentially no precedent for computing functions as complex as these [34–36]. We also note that the structure of our CRNs is uniform and "feed-forward": the output species are computed directly from the input species, with no coupling or complex feedback dynamics. Accordingly, the computation should be highly accurate and robust.

A significant feature of our design is that the encoding of the outputs is the same as that of the inputs. The output of each CRN is encoded by a pair of molecular species, say $C_0$ and $C_1$, whose relative concentration encodes a stochastic value, $c = C_1/(C_0 + C_1)$. This is exactly the same format as the inputs, say $a = A_1/(A_0 + A_1)$, and $b = B_1/(B_0 + B_1)$. Therefore the output from a CRN can be used as the input to another CRN.

The volume of all input solutions can be scaled up to allow the production of more output solution. This allows for "fanout": dividing the output solution into multiple parts each used as inputs to other CRNs. For example, if the output of a CRN feeds into four subsequent CRNs, its inputs must be scaled up by a factor of 4. Its output can then be volumetrically split into four separate units that can be fed into each of the subsequent CRNs.

## 4 Proof of the proposed method

Here we prove the correctness of the method of implementing stochastic functions with CRNs discussed in Section 3. We then elucidate the proof with a simple example in Section 4.1. (Readers may want to step through this example first and then return to the proof.)

**Theorem 2** *Assume an n-input stochastic function $y = f(x_1, x_2, \ldots, x_n)$ is implemented by a combinational Boolean function $Y = F(X_1, X_2, \ldots, X_n)$. The stochastic function can then be implemented with a CRN with $2n + 2$ different molecular species, in which pairs of molecular species store the input values $x_1, x_2, \ldots, x_n$ as well as the output value $y$, according to the fractional representation in* Eq 13. *The CRN consists of $2^n$ reactions, each of the form,*

$$\mathbf{X}_{1,v_1} + \mathbf{X}_{2,v_2} + \ldots + \mathbf{X}_{n,v_n} \xrightarrow{k} \mathbf{Y}_{F(V)}, \tag{17}$$

*where $v_1, v_2, \ldots, v_n$: $F(V)$ is a row of the truth table for the combinational function F, and $V = (v_1, v_2, \ldots v_n)$ denotes a minterm for the function. Note that the rate constants for all reactions are equal to k, an arbitrary value.*

Let $S_1$ be the set of all minterms $V$ such that $F(V) = 1$, and let $S_0$ be the set of all minterms $V$ such that $F(V) = 0$. Also, we denote $c_{i,j}$ as,

$$c_{i,j} = \Pr(X_i = j) = \begin{cases} 1 - x_i & \text{if } j = 0 \\ x_i & \text{if } j = 1 \end{cases} \tag{18}$$

where $x_i$ is a stochastic input, and $i$ is the index of the input $x_i$ in function $y = f(x_1, x_2, \ldots, x_n)$.

To prove the theorem, we need to show that, for the given initial values of the stochastic value $x_i$ at time $t = 0$,

$$x_i = \left. \frac{[\mathbf{X}_{i,1}]}{[\mathbf{X}_{i,0}] + [\mathbf{X}_{i,1}]} \right|_{t=0}, \tag{19}$$

the output of the CRN should match the output of the stochastic function stated in Theorem 1,

$$\lim_{t \to \infty} y = \lim_{t \to \infty} \frac{[\mathbf{Y}_1]}{[\mathbf{Y}_0] + [\mathbf{Y}_1]} = \sum_{V \in S_1} \left( \prod_{h=1}^{n} c_{h,v_h} \right). \tag{20}$$

In fact, we prove an even stronger result that the limit $t \to \infty$ in Eq 20 is not necessary: that is, at any $t > 0$

$$y = \frac{[\mathbf{Y}_1]}{[\mathbf{Y}_0] + [\mathbf{Y}_1]}. \tag{21}$$

*Proof* Given the CRN described in Theorem 4, the rate equations for each input are

$$\frac{d[\mathbf{X}_{i,j}]}{dt} = -k \cdot [\mathbf{X}_{i,j}] \cdot \prod_{h=1,h\neq i}^{n} ([\mathbf{X}_{h,0}] + [\mathbf{X}_{h,1}]), j \in \{0, 1\} \tag{22}$$

$$= -k \cdot \frac{[\mathbf{X}_{i,j}]}{[\mathbf{X}_{i,0}] + [\mathbf{X}_{i,1}]} \cdot \prod_{h=1}^{n} ([\mathbf{X}_{h,0}] + [\mathbf{X}_{h,1}]). \tag{23}$$

Note that $k$, an arbitrary value, is the rate constant for each reaction. The rate equations for the output species are,

$$\frac{d[\mathbf{Y}_j]}{dt} = k \sum_{V \in S_j} \left( \prod_{h=1}^{n} [\mathbf{X}_{h,v_h}] \right), j \in \{0, 1\}. \tag{24}$$

We define the following new variables,

$$p_i = \frac{[\mathbf{X}_{i,1}]}{[\mathbf{X}_{i,0}] + [\mathbf{X}_{i,1}]} \tag{25}$$

$$q_i = [\mathbf{X}_{i,0}] + [\mathbf{X}_{i,1}] \tag{26}$$

$$r_{i,j} = \begin{cases} 1 - p_i & \text{if } j = 0 \\ p_i & \text{if } j = 1 \end{cases} \tag{27}$$

We substitute these variables into the expressions for the concentrations:

$$[\mathbf{X}_{i,0}] = q_i(1 - p_i), \tag{28}$$

$$[\mathbf{X}_{i,1}] = q_i p_i, \tag{29}$$

$$\text{Therefore, } [\mathbf{X}_{i,j}] = q_i r_{i,j}. \tag{30}$$

These substitutions are introduced into Eqs 23 and 24:

$$\frac{d[\mathbf{X}_{i,j}]}{dt} = -k \cdot r_{i,j} \prod_{h=1}^{n} q_h, \tag{31}$$

$$\frac{d[\mathbf{Y}_j]}{dt} = k \left( \prod_{h=1}^{n} q_h \right) \sum_{V \in S_j} \left( \prod_{h=1}^{n} r_{h,v_h} \right). \tag{32}$$

As the concentrations $[\mathbf{X}_{i,j}]$ are functions of time, all $p$, $q$, and $r$ are also functions of time. Consider the following two expressions derived from Eq 31,

$$\frac{d[\mathbf{X}_{i,0}]}{dt} = -k(1 - p_i) \prod_{h=1}^{n} q_h \tag{33}$$

$$\frac{d[\mathbf{X}_{i,1}]}{dt} = -k \cdot p_i \prod_{h=1}^{n} q_h. \tag{34}$$

$$\text{Therefore, } \frac{dq_i}{dt} = \frac{d[\mathbf{X}_{i,0}]}{dt} + \frac{d[\mathbf{X}_{i,1}]}{dt} = -k \prod_{h=1}^{n} q_h. \tag{35}$$

We also have

$$[\mathbf{X}_{i,1}] \quad = \quad p_i \cdot q_i \tag{36}$$

Therefore, $\dfrac{d[\mathbf{X}_{i,1}]}{dt} \quad = \quad p_i \dfrac{dq_i}{dt} + q_i \dfrac{dp_i}{dt}$ (37)

$$= \quad p_i\left(-k\prod_{h=1}^{n}q_i\right) + q_i\dfrac{dp_i}{dt} \tag{39}$$

$$= \quad \dfrac{d[\mathbf{X}_{i,1}]}{dt} + q_i\dfrac{dp_i}{dt}. \tag{39}$$

As $q_i \neq 0$, we conclude that

$$\dfrac{dp_i}{dt} = 0, \tag{40}$$

that is, $p_i$ is invariant to time. Consequently, $r_{i,j}$ is also invariant to time. This means that the stochastic value encoded by each pair of input species remains the same throughout the reaction. Therefore, for $t > 0$, we have

$$p_i \quad = \quad x_i \tag{41}$$

$$r_{i,j} \quad = \quad c_{i,j} \tag{42}$$

We assign the new symbol

$$l \quad = \quad \left(\prod_{h=1}^{n}q_i\right) \tag{43}$$

Therefore, $\dfrac{d[\mathbf{Y}_j]}{dt} \quad = \quad k \cdot l \displaystyle\sum_{V \in S_j}\left(\prod_{h=1}^{n}r_{h,v_h}\right).$ (44)

Finally, we can calculate the stochastic output $y$ as

$$y \quad = \quad \dfrac{\displaystyle\int_{0}^{t}\dfrac{d[\mathbf{Y}_1]}{dt}dt}{\displaystyle\int_{0}^{t}\dfrac{d[\mathbf{Y}_0]}{dt}dt + \int_{0}^{t}\dfrac{d[\mathbf{Y}_1]}{dt}dt} \tag{45}$$

$$= \frac{\sum_{V \in S_1} \left( \prod_{h=1}^{n} r_{h,v_h} \right) \int_0^t k \cdot l \cdot dt}{\sum_{V \in S_0} \left( \prod_{h=1}^{n} r_{h,v_h} \right) \int_0^t k \cdot l \cdot dt + \sum_{V \in S_1} \left( \prod_{h=1}^{n} r_{h,v_h} \right) \int_0^t k \cdot l \cdot dt} \tag{46}$$

$$= \frac{\sum_{V \in S_1} \left( \prod_{h=1}^{n} r_{h,v_h} \right)}{\sum_{V \in S_0} \left( \prod_{h=1}^{n} r_{h,v_h} \right) + \sum_{V \in S_1} \left( \prod_{h=1}^{n} r_{h,v_h} \right)} \tag{47}$$

$$= \sum_{V \in S_1} \left( \prod_{h=1}^{n} r_{h,v_h} \right). \tag{48}$$

The numerator in Eq 47 corresponds to the sum of the minterms of all rows of the truth table $F$ that evaluate 1, while the denominator corresponds to the sum of all minterms. As $r_{i,j}$ is only dependent on the initial input value, the denominator must sum up to 1 since it includes all the minterms. Therefore, we conclude that a CRN constructed this way, corresponding to an arbitrary Boolean truth table $F$, will implement the stochastic function $f$ of that truth table. The only requirement is that the rate constants of all the reactions must be equal.

In what follows, we elucidate the proof with an example. In the Supporting Information, we give CRN implementations of a variety of functions that are of practical interest.

## 4.1 A demonstrative example

Let us go back to the two-input AND gate from Section 3.

$$\mathbf{A}_0 + \mathbf{B}_0 \xrightarrow{k} \mathbf{C}_0$$

$$\mathbf{A}_0 + \mathbf{B}_1 \xrightarrow{k} \mathbf{C}_0$$

$$\mathbf{A}_1 + \mathbf{B}_0 \xrightarrow{k} \mathbf{C}_0 \tag{49}$$

$$\mathbf{A}_1 + \mathbf{B}_1 \xrightarrow{k} \mathbf{C}_1$$

The rate equations for the input and output species are:

$$\frac{d[\mathbf{A}_0]}{dt} = -k[\mathbf{A}_0]([\mathbf{B}_0] + [\mathbf{B}_1])$$

$$\frac{d[\mathbf{A}_1]}{dt} = -k[\mathbf{A}_1]([\mathbf{B}_0] + [\mathbf{B}_1])$$

$$\frac{d[\mathbf{B}_0]}{dt} = -k[\mathbf{B}_0]([\mathbf{A}_0] + [\mathbf{A}_1])$$

$$\frac{d[\mathbf{B}_1]}{dt} = -k[\mathbf{B}_1]([\mathbf{A}_0] + [\mathbf{A}_1]) \tag{50}$$

$$\frac{d[\mathbf{C}_0]}{dt} = k([\mathbf{A}_0][\mathbf{B}_0] + [\mathbf{A}_0][\mathbf{B}_1] + [\mathbf{A}_1][\mathbf{B}_0])$$

$$\frac{d[\mathbf{C}_1]}{dt} = k[\mathbf{A}_1][\mathbf{B}_1].$$

We introduce some variables to represent the stochastic values,

$$a = \frac{[\mathbf{A}_1]}{[\mathbf{A}_0] + [\mathbf{A}_1]}, \quad b = \frac{[\mathbf{B}_1]}{[\mathbf{B}_0] + [\mathbf{B}_1]}, \quad c = \frac{[\mathbf{C}_1]}{[\mathbf{C}_0] + [\mathbf{C}_1]},$$

as well as the sum of concentrations of each pair of input species,

$$[\mathbf{A}_0] + [\mathbf{A}_1] = q_a, \quad [\mathbf{B}_0] + [\mathbf{B}_1] = q_b.$$

With these variables, Eq 50 becomes:

$$\frac{d[\mathbf{A}_0]}{dt} = -kq_a q_b \cdot (1 - a)$$

$$\frac{d[\mathbf{A}_1]}{dt} = -kq_a q_b \cdot a$$

$$\frac{d[\mathbf{B}_0]}{dt} = -kq_a q_b \cdot (1 - b)$$

$$\frac{d[\mathbf{B}_1]}{dt} = -kq_a q_b \cdot b \tag{51}$$

$$\frac{d[\mathbf{C}_0]}{dt} = kq_a q_b \cdot [(1 - a)(1 - b) + (1 - a)b + a(1 - b)]$$

$$\frac{d[\mathbf{C}_1]}{dt} = kq_a q_b \cdot ab.$$

Let us prove the time invariance of $a$ and $b$. We can express $[\mathbf{A}_1]$ as $a \cdot q_a$, therefore according to the chain rule for derivatives,

$$\frac{d[\mathbf{A}_1]}{dt} = q_a \frac{da}{dt} + a \frac{dq_a}{dt}. \tag{52}$$

According to Eq 51,

$$\frac{dq_a}{dt} = \frac{d[\mathbf{A}_1]}{dt} + \frac{d[\mathbf{A}_1]}{dt} = -kq_a q_b. \tag{53}$$

From Eqs 51, 52 and 53, we conclude that,

$$q_a \frac{da}{dt} = 0. \tag{54}$$

Since, during the process, $q_a$ is not a constant equal to 0, we conclude that $\frac{da}{dt} = 0$. This proves the time invariance of $a$, that is to say, during the process, the fractional value encoded by $[\mathbf{A}_0]$ and $[\mathbf{A}_1]$ remains the same. Similarly, we can prove that $b$ is time-invariant.

From here, we can calculate $c$ for $t > 0$. Assume the initial concentration of $[\mathbf{C}_0]$ and $[\mathbf{C}_1]$ are 0, then

$$
\begin{aligned}
c \quad &= \frac{[\mathbf{C}_1]}{[\mathbf{C}_0] + [\mathbf{C}_1]} \\[2ex]
&= \frac{\displaystyle\int_0^t \frac{d[\mathbf{C}_1]}{dt} dt}{\displaystyle\int_0^t \frac{d[\mathbf{C}_0]}{dt} dt + \int_0^t \frac{d[\mathbf{C}_1]}{dt} dt} \\[2ex]
&= \frac{\displaystyle\int_0^t kq_a q_b \cdot ab \cdot dt}{\displaystyle\int_0^t kq_a q_b dt} \\[2ex]
&= \frac{ab \displaystyle\int_0^t kq_a q_b dt}{\displaystyle\int_0^t kq_a q_b dt} \quad (\text{since } a, b \text{ are constant}) \\[2ex]
&= ab.
\end{aligned}
\tag{55}
$$

This proves that an AND gate implements multiplication.

## 5 Error analysis

We performed simulations to test the robustness of CRNs implementing stochastic functions with the program *Mathematica* [37]. The code is given in S2 File. ManuscriptWe generated differential equations corresponding to the reaction kinetics for CRNs and investigated the impact of varying reaction rates. Here we present a detailed analysis for a specific CRN, one that implements the polynomial:

$$f(x, y, z) = x + y + z - 2xy - 2xz - 2yz + 4xyz. \tag{56}$$

We deliberately chose this function, a 3-input Exclusive-OR (XOR), as our error case because the truth table for XOR is balanced in terms of the number of 0's and 1's. Accordingly, it is the most sensitive to random variations in reaction rates. In contrast, for unbalanced functions such as AND or OR, errors can readily be masked: computing more 0's for AND or more 1's for OR may not show up statistically.

This polynomial for this function is generated by the following truth table:

| $x$ | $y$ | $z$ | $f(x, y, z)$ |
|---|---|---|---|
| 0 | 0 | 0 | 0 |
| 0 | 0 | 1 | 1 |
| 0 | 1 | 0 | 1 |
| 0 | 1 | 1 | 0 |
| 1 | 0 | 0 | 1 |
| 1 | 0 | 1 | 0 |
| 1 | 1 | 0 | 0 |
| 1 | 1 | 1 | 1 |

To see this, take the sum of the expressions for the minterms, i.e., the rows that evaluate to one. Recall that the expression for each row is formed by multiplying together factors corresponding to the input variables: $x$ if the variable $x$ is equal to 1 or $1 - x$ if the variable $x$ is equal to 0:

$$
\begin{aligned}
f(x, y, z) &= (1 - x)(1 - y)z + \\
&\quad (1 - x)y(1 - z) + \\
&\quad (1 - x)y(1 - z) + \\
&\quad x(1 - y)(1 - z) + \\
&\quad xyz \\
&= x + y + z - 2xy - 2xz - 2yz + 4xyz.
\end{aligned}
\tag{57}
$$

According to the method discussed in Section 3, we can translate this truth table into a CRN as follows:

$$
\begin{aligned}
\mathbf{X}_0 + \mathbf{Y}_0 + \mathbf{Z}_0 &\xrightarrow{k_1} \mathbf{F}_0 \\
\mathbf{X}_0 + \mathbf{Y}_0 + \mathbf{Z}_1 &\xrightarrow{k_2} \mathbf{F}_1 \\
\mathbf{X}_0 + \mathbf{Y}_1 + \mathbf{Z}_0 &\xrightarrow{k_3} \mathbf{F}_1 \\
\mathbf{X}_0 + \mathbf{Y}_1 + \mathbf{Z}_1 &\xrightarrow{k_4} \mathbf{F}_0 \\
\mathbf{X}_1 + \mathbf{Y}_0 + \mathbf{Z}_0 &\xrightarrow{k_5} \mathbf{F}_1 \\
\mathbf{X}_1 + \mathbf{Y}_0 + \mathbf{Z}_1 &\xrightarrow{k_6} \mathbf{F}_0 \\
\mathbf{X}_1 + \mathbf{Y}_1 + \mathbf{Z}_0 &\xrightarrow{k_7} \mathbf{F}_0 \\
\mathbf{X}_1 + \mathbf{Y}_1 + \mathbf{Z}_1 &\xrightarrow{k_8} \mathbf{F}_1
\end{aligned}
\tag{58}
$$

Note that the indices of the molecular species match the entries in the truth table above. Since we will be exploring the consequences of non-uniform rate constants, note that here we have assigned the eight reactions unique rate constants: $k_1, k_2, \ldots k_8$, respectively. We can verify that this CRN implements the function in Eq 56 through the differential equations. We define

the following stochastic variables:

$$x = \frac{[\mathbf{X}_1]}{[\mathbf{X}_0] + [\mathbf{X}_1]}, \quad y = \frac{[\mathbf{Y}_1]}{[\mathbf{Y}_0] + [\mathbf{Y}_1]}, \quad z = \frac{[\mathbf{Z}_1]}{[\mathbf{Z}_0] + [\mathbf{Z}_1]}, \quad f = \frac{[\mathbf{F}_1]}{[\mathbf{F}_0] + [\mathbf{F}_1]}. \tag{59}$$

We used the procedure `NDSolveValue` in *Mathematica* to simulate the differential equations corresponding to CRN in Eq 6. We varied the rate constants as well as the initial concentrations. We compared the value of $f$ computed by the CRN, in terms of $[F_0]$, $[F_1]$ to the expected value of $f$ from Eq 56. Here is a summary of the trials:

### 5.1 Trials for error analysis

The error was calculated as the absolute difference between the value computed by the CRN simulation and the expected value of $f$ from Eq 56.

1. With all $k_i$ = 100 except for $k_1$ = 1000, i.e., one rate constant being an order of magnitude higher than the others: the highest error observed was 0.31, with 38.1% of the input combinations having an error greater than 0.1.

2. With all $k_i$ = 100 except for $k_1$ = 10, i.e., one rate constant being an order of magnitude lower than the others: the highest error observed was 0.12, with 15.7% of the input combinations having an error greater than 0.1.

3. With all $k_i$ = 100 except for $k_1$ = 10000, i.e., one rate constant being two orders of magnitude higher than the others: the highest error observed was 0.45, with 45.8% of the input combinations having an error greater than 0.1.

4. With all $k_i$ = 100 except for $k_1$ = 1, i.e., one rate being two orders of magnitude lower than the others: the highest error recorded was 0.12, with 22.7% of the input combinations having an error greater than 0.1.

5. With all $k_i$ randomly generated, from a normal distribution with a mean of 100 and a low standard deviation of 10: the highest error recorded was 0.06, with no input combinations having an error greater than 0.1.

6. With all $k_i$ randomly generated, from a normal distribution with a mean of 100 and a high standard deviation of 70 (negative values were not allowed): the highest error recorded was 0.25, with 14.4% of the input combinations having an error greater than 0.1.

The absolute difference between the output value of $f$, calculated with Eq 59, compared to the expected value of $f$ from Eq 56 was calculated for a wide range of input concentrations. These are graphed in Fig 2. The inputs $x$, $y$, and $z$, calculated with Eq 59, were set to values in the interval [0, 1] forming a cube mesh input. All input chemical species were initialized such that $[\mathbf{X}_0] + [\mathbf{X}_1]$ = 100. The maximum error difference and the number of input combinations for which the error differential exceeded 0.1 were recorded. The purpose of this simulation was not to account for all possible values of the rate constants, but rather to understand the design constraints and the error margins. The key observations from our simulations are:

1. In a network with many reactions, one rate constant being slower than the others by an order of magnitude or two has a lower impact on error than if it were faster by a similar amount.

2. Error rates are low if all the rate constants are within the same order of magnitude and are distributed normally with a small standard deviation.

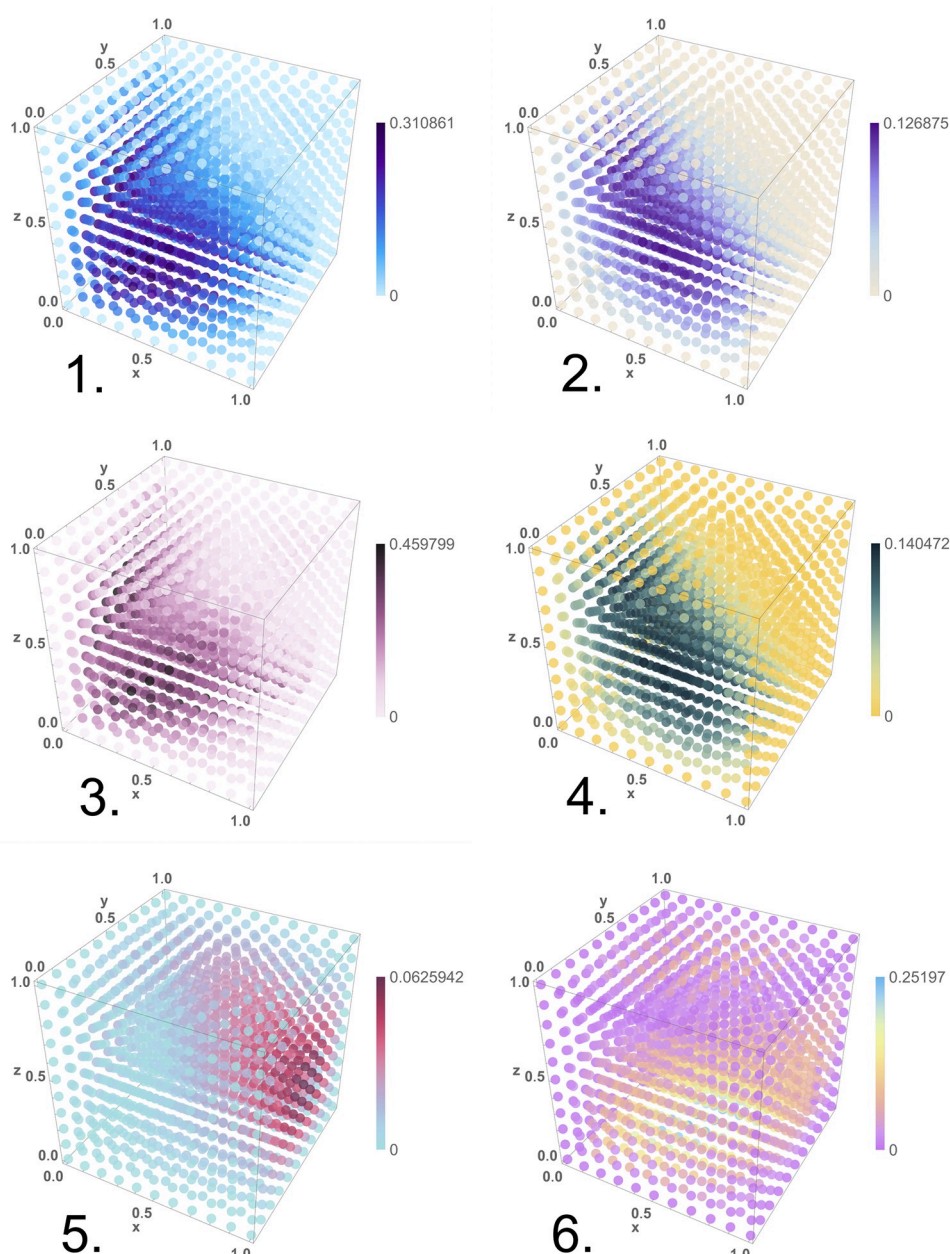

**Fig 2. The error cubes for the six trials listed in Section 5.1.** The three dimensions in the plots span the inputs $x$, $y$, and $z$, each in the interval $[0, 1]$, with a step size 0.1. The color of each point corresponds to the absolute difference between the value computed by the CRN and the expected value of $f$ from Eq 56. A legend is provided for each cube. The trials were performed with theNDSolveValue function in software tool *Mathematica*.

3. Error rates are also low when some of the fractional inputs are close to 0 or to 1. This translates to very slow or very fast reactions, respectively.

4. Even when the rate constants differ by orders of magnitude, not all inputs result in high errors. Simulation is a valuable guide.

## 6 Implementation using DNA

### 6.1 DNA strand-displacement

DNA strand displacement is a well-established technique for implementing molecular computation [38, 39]. Prior work has shown that such a system can emulate *any* abstract set of chemical reactions. The reader is referred to Soloveichik et al. and Zhang et al. for further details [18, 40]. Here we illustrate a simple, generic example. In Section 6.2, we discuss how to map our models to such DNA strand-displacement systems.

We begin by first defining a few basic concepts. DNA strands are linear sequences of four different nucleotides {*A*, *T*, *C*, *G*}. A nucleotide can bind to another following *Watson-Crick* base-pairing: A binds to T, C binds to G. A pair of single DNA strands will bind to each other, a process called *hybridization*, if their sequences are complementary according to the base-pairing rule, that is to say, wherever there is an *A* in one, there is a *T* in the other, and vice versa; and whenever there is a *C* in one, there is a *G* in the other and vice-versa. The binding strength depends on the length of the complementary regions. Longer regions will bind strongly, smaller ones weakly. Reaction rates match binding strength: hybridization completes quickly if the complementary regions are long and slowly if they are short. If the complementary regions are very short, hybridization might not occur at all. (We acknowledge that, in this brief discussion, we are omitting many relevant details such as temperature, concentration, and the distribution of nucleotide types, i.e., the fraction of paired bases that are A-T versus C-G. All of these parameters must be accounted for in realistic simulation runs.)

Fig 3 illustrates strand displacement with a set of reversible reactions. The entire reaction occurs as reactant molecules *A* and *B* form products *E* and *F*, with each intermediate stage operating on molecules *C* and *D*. In the figure, *A* and *F* are single strands of DNA, while *B*, *C*, *D*, and *E* are double-stranded complexes. Each single-strand DNA molecule is divided, conceptually, into subsequences that we call **domains**, denoted as 1, 2, and 3 in the figure. The complementary sequences for these domains are 1*, 2* and 3*. (We will use this notation for complementarity throughout.) All distinct domains are assumed to be *orthogonal* to each other, meaning that these domains do not hybridize.

**Toeholds** are a specific kind of domain in a double-stranded DNA complex where a single strand is exposed. For instance, the molecule *B* contains a toehold domain at 1* in Fig 3. Toeholds are usually 6 to 10 nucleotides long, while the lengths of regular domains are typically 20 nucleotides. The exposed strand of a toehold domain can bind to the complementary domain from a longer single DNA strand, and thus toeholds can trigger the binding and displacement of DNA strands. The small length of the toehold makes this hybridization reversible.

In the first reaction in Fig 3, the open toehold 1* in molecule *B* binds with domain 1 from strand *A*. This forms the molecule *C* where the duplicate 2 domain section from molecule *A* forms an overhanging flap. This reaction shows how a toehold triggers the binding of DNA strands. In molecule *C*, the overhanging flap can stick onto the complementary domain 2*, thus displacing the previously bound strand. This type of branch migration is shown in the second reaction, where the displacement of one flap to the other forms the molecule *D*. This reaction is reversible, and the molecules *C* and *D* exist in a dynamic equilibrium. The process of branch migration of the flap is essentially a random walk: at any time when part of the strand from molecule *A* hybridizes with strand *B*, more of *A* might bind and displace a part of *F*, or more of *F* might bind and displace a part of *A*. Therefore, this reaction is reversible. The third reaction is the exact opposite of reaction 1—the new flap in molecule *D* can peel off from the complex and thus create the single-strand molecule *F* and leave a new double-stranded complex *E*. Molecule *E* is similar to molecule *B*, but the toehold has migrated from 1* to 3*. The reaction rate of this reaction depends on the length of the toehold 3*. If we reduce the

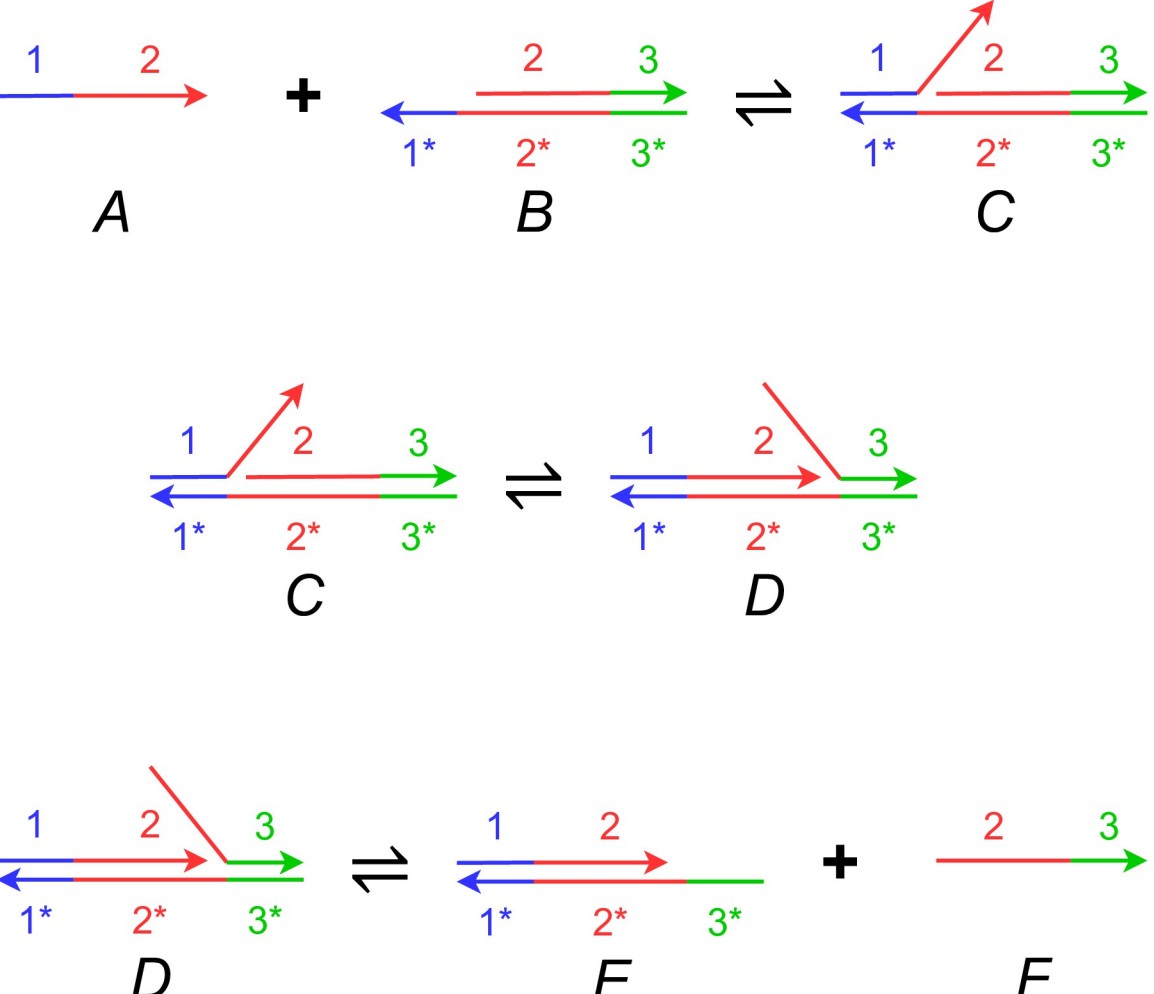

**Fig 3. A set of DNA strand displacement reactions.** Each DNA single strand is drawn as a continuous arrow, consisting of different colored domains numbered 1 through 3. DNA domains that are complementary to each other due to A-T, C-G binding are paired as 1 and 1*. The first reaction shows reactants A and B hybridizing together via the toehold at domain 1* on molecule B. The second reaction depicts branch migration of the overhanging flap of DNA in molecule C, thereby resulting in the nick migrating from after domain 1 to 2. The third reaction shows how an overhanging strand of DNA can be peeled off of molecule D, thereby exposing a toehold at domain 3* on molecule E and releasing a freely floating strand F. All reactions are reversible. The only domains that are toeholds are 1* and 3*.

length of the toehold, the rate of reaction 3 becomes so small that the reaction can be treated as a forward-only reaction. This bias in the direction of the reaction means that we can model the entire set of reactions as a single DNA strand displacement event, where reactants *A* and *B* react to produce *E* and *F*. Note that the strand *F* can now participate in further toehold-mediated reactions, allowing for cascading of such these DNA strand displacement systems.

## 6.2 DNA concatemers

DNA Concatemers are long strands of DNA that contain repeated base-pair sequences. These are formed when a single smaller DNA unit is capable of hybridizing with other copies of itself. Specifically, to form a DNA strand of the form **A B A B A B**. . ., the 1-mer unit must have the following 3 regions:

1. A leading sticky end (single-stranded region) on the 1st strand with the sequence **A**.

2. A middle double-stranded section with the sequence **B**.

3. A trailing sticky end on the 2nd strand with the complement sequence $\mathbf{A}'$ such that it can bind to a leading sticky end for **A**.

We propose designing our molecules for fractional representation as DNA concatemers [41] that can interact via strand displacement, as detailed in the next subsection. For a fractional variable $a$, the molecules $\mathbf{A}_0$ and $\mathbf{A}_1$ needed for the reaction network can be designed as concatemer units such that the double-stranded section for each unit is distinct, but the sticky ends for both of them are the same. This allows the two species to cross-polymerize and forms a linear chain of DNA of randomly arranged $\mathbf{A}_0$ and $\mathbf{A}_1$ units. This is similar to the randomized digital bitstreams used in stochastic computing in which a random stream of 0's and 1's forms the basic data unit [3, 13]: $\mathbf{A}_0$ and $\mathbf{A}_1$ correspond to 0 and 1, respectively. Thus a single fractional variable can be stored as a long DNA strand that can be amplified to improve readout [42]; this long strand can then be broken up using artificial restriction enzymes—or natural restriction enzymes, if the sticky ends are designed purposefully. Furthermore, this concatemer design allows the use of RNA-seq [43] in the readout process to measure the fractional value stored by a DNA strand. For this purpose, a long DNA concatemer must be broken into its constituent monomers using a restriction enzyme, and then these smaller DNA units can be used instead of the standard complementary DNA in RNA-seq to determine the expression level of each unit. From this quantitative readout, the relative amount of $\mathbf{A}_1$ to $\mathbf{A}_0 +$ $\mathbf{A}_1$ can be determined [44].

## 6.3 Procedure

Fig 4 illustrates the reaction $\mathbf{A}_i + \mathbf{B}_j \rightarrow \mathbf{C}_k$ implemented with DNA strand displacement and cleaving enzymes. Two species of concatemer units are transformed into another concatemer unit. The implementation consists of three stages:

1. Extracting single strands: Consider the two input concatemers $\mathbf{A}_i$ and $\mathbf{B}_j$ shown in the figure. We design the concatemers in such a way that the sticky ends of a concatemer unit can act like open toeholds in DNA strand displacement. As a result, we can extract a single strand from a concatemer. For example, concatemer $\mathbf{A}_i$ is formed with two single strands $[T_i, A_i]$, $[A_i^*, T_1^*]$. We can add strand $[A_i, T_1]$ so that strand $[T_1, A_i]$ is displaced. Similarly, we can extract strand $[T_2, B_j, T_3]$ from concatemer $\mathbf{B}_j$ with strand $[B_j, T_3, T_2]$.

2. This is the strand displacement reaction that implements the main reaction. It receives two single-strand DNA molecules, $[T_1, A_i]$ and $[T_2, B_j, T_3]$ as reactants. The product is a complex containing the output concatemer. The reaction is divided into two parts. In the first part, strand $[T_1, A_i]$ displaces strand $[A_i, T_2]$ from the auxiliary complex $\mathbf{G}_1$ and forms $\mathbf{G}_2$ through a reversible reaction. Then the strand $[T_2, B_j, T_3]$ displaces the output complex which is formed by strand $[B_j, T_3, C_k]$ and $[C_k^*, T_3^*]$. This step is irreversible since the output complex cannot bind to the resulting auxiliary complex $\mathbf{G}_3$ after this step.

3. Cleaving. The output complex from the previous step contains the domain $B_j$ in addition to the part that could form concatemer $\mathbf{C}_k$. The domain $B_j$ is cleaved from the complex. After this step, we get a concatemer $\mathbf{C}_k$ with $T_3$ sticky end. Cleaving can be achieved by using DNA editing enzymes such as CRISPR-Cas9 and PfAgo [45].

We assume that the concentration of the initial auxiliary complex $\mathbf{G}_1$ is much larger than the concentration of the concatemers. With this assumption, the concentration of the auxiliary

## Input Concatemers

## 1. Extract Single Strand

## 2. Reaction Step

## 3. Cleave

**Fig 4. An example illustrating strand displacement reactions, implemented using concatemers.** The figure is divided into an example sequence of concatemers, and three reaction steps: 1) extracting a single strand from concatemers; 2) a reaction step that consumes two single strands and outputs a complex; and 3) cleaving.

complex can be treated invariant through the reaction. Thus, the reaction rate only depends on the concentration of the single strands extracted from the concatemers. As there are four reactions to implement the two-input network shown in this example, four species of the auxiliary complex representing each reaction should be used. This ensures that the mixture of different species of $A_0$ and $A_1$, or $B_0$ and $B_1$, can react competitively. During the cleaving step, each reactant participates in only one reaction. Therefore, it should not affect the reaction rate or the fractional encoding of the output by the two product species.

The reaction itself can be extended to a multimolecular reaction by extending the chain of toehold exchange reactions. Suppose, for example, a new stochastic value $d$ with molecules $D_l$ and sticky ends $T_4$ were also the input alongside $a$ and $b$. In the complex $G_1$, domains $[T_4, D_l]$ and their complementary domains would be added between the domains $B_j$ and $T_3$. That is, a new $G_1$ that would react with single strands of sequence $D_l$ and toehold $T_4$ would be used. In this way, $G_1$ would be capable of receiving an additional strand $[T_4, D_l]$ before displacing the final product. Therefore multiple input values can be computed upon in our CRNs.

When computing with digital circuits, the length of the bitstream dictates the precision of the computation. The length of the bitstream can be chosen by the user based on their specifications. The more precision that they require, the longer the bitstream that they should use. In our DNA implementation, the concentration of DNA concatemers corresponds to the length of the bitstreams for the stochastic functions. So the limitation is experimental: how precisely the user can set and measure the input and output concentrations, respectively.

## 7 Conclusion

This paper proposed a strategy for computing mathematical functions with molecular systems based on a fractional representation, using a pair of molecular species to represent each mathematical variable. With this representation, we can apply the theory of stochastic logic design chemical reaction networks for computing functions. In particular, we showed how to translate the truth tables for stochastic functions into chemical reaction networks. We then demonstrated how to implement the reaction networks with DNA strand displacement.

Stochastic logic is an intriguing paradigm for digital computation. Instead of computing definite outputs from definite inputs—say Boolean values from Boolean values, or integers from integers—it entails computing probabilities from probabilities. There is randomness and yet the computation is robust. The computation is effected by transforming the *statistical distribution* of random bitstreams. The paradigm has been applied in a variety of domains, particularly for emerging technologies [2, 46–48]. It has been most successful for applications that entail computing mathematical functions: for instance, `arctan` for nonlinear activation functions in machine learning; *Bessel* functions for differential system models; and the `sinc` function for image and signal processing. We give examples of CRN implementations of these functions in the Supporting Information. Of course, we cannot point to real-world applications that call for the molecular computation of such functions. For now, the ideas in this paper should be taken as a proof of concept.

Over the past two decades, computing has moved from desktops and data centers into the wild. Embedded microchips—found in our gadgets, our tools, our buildings, our soils and even our bodies—are transforming our lives. And yet, there are limits to where silicon can go and where it can compute effectively. It operates based on voltage and so requires a power source. Even miniaturized to the microscale or smaller, an electronic system is often a foreign object inserted into a material, substrate, or environment. This sort of computation discussed in this paper could find application in a novel class of computing system that is not foreign, but rather an integral part of its physical and chemical environment: a system that computes

*with* its constituent molecules. In such a system, sensing, computing, and actuating occur at the molecular level, with no interfacing at all with external electronics. Futuristic, yes, but we can point to the field of *soft robotics* where such systems are being developed [49].

## Supporting information

**S1 File. Examples of CRNs for polynomial approximations of nonlinear functions.** We calculate the CRN for polynomial approximations of various functions such as ArcTan, Exponential, Bessel, and Sinc.
(PDF)

**S2 File. Mathematica script for the error analysis of the 3-input XOR.** We use the `NDSolveValue` command in *Mathematica* to simulate the system of differential equations for the 3-input XOR in Section 5. The script includes the various error analyses and image printing commands.
(PDF)

## Acknowledgments

We thank David Soloveichik, Olgica Milenkovic, Andrew Ellington for helpful discussions. In particularly, we thank Andrew Ellington for suggesting that we use DNA "concatemers."

## Author Contributions

**Conceptualization:** Marc Riedel.

**Formal analysis:** Tonglin Chen, Marc Riedel.

**Investigation:** Marc Riedel.

**Software:** Arnav Solanki.

**Supervision:** Marc Riedel.

**Visualization:** Arnav Solanki.

**Writing – original draft:** Arnav Solanki, Tonglin Chen, Marc Riedel.

**Writing – review & editing:** Marc Riedel.

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
