## [Decision Letter · Decision Letter 0]

21 Nov 2022

PONE-D-22-24554Computing Mathematical Functions with Chemical Reactions via Stochastic Logic

PLOS ONE

Dear Dr. Riedel,

Thank you for submitting your manuscript to PLOS ONE. Your manuscript has been seen by two reviewers. Both reviews are curious about feasibility of implementation and wonder what technical limitations would be imposed by a real molecule in terms of, e.g., the size of the code. I agree with these questions. We invite you to submit a revised version of the manuscript that addresses the points raised during the review process. 

We look forward to receiving your revised manuscript.

Kind regards,

Ivan Kryven

Academic Editor

PLOS ONE

Journal Requirements:

“This work was funded by DARPA Grant #W911NF-18-2-0032.”

“MR

DARPA Grant W911NF-18-2-0032 https://www.darpa.mil

4. Please ensure that you refer to Figure 1 in your text as, if accepted, production will need this reference to link the reader to the figure.

Reviewers' comments:

Reviewer's Responses to Questions

**Comments to the Author**

1. Is the manuscript technically sound, and do the data support the conclusions?

Reviewer #1: Yes

Reviewer #2: Yes

2. Has the statistical analysis been performed appropriately and rigorously? 

Reviewer #1: Yes

Reviewer #2: N/A

3. Have the authors made all data underlying the findings in their manuscript fully available?

Reviewer #1: Yes

Reviewer #2: Yes

4. Is the manuscript presented in an intelligible fashion and written in standard English?

Reviewer #1: Yes

Reviewer #2: Yes

5. Review Comments to the Author

Reviewer #1: The first part of the paper recalls results in stochastic logic on

using logic gates to operate over random bit streams as encodings of

real numbers and compute polynomials; and it exhibits a procedure for

chemical reaction network construction that results in CRNs that

faithfully implement such logic gates and thus correctly compute

desired polynomials. A beautiful result proven here is that the

numerical intepretation of the inputs is not perturbed by the

reaction, and that the numerical interpretation of the outputs is

correct immediately. I didn't anticipate this outcome; it is very

nice.

The second part, Section 7, proposes a way to construct long

structured DNA strands that could be a physical representation of

random bit streams, and ways to access individual monomers for the

purpose of logic operations. As far as proposals go, this one is

plausible, but it is impossible to foresee all the difficulties of a

wet lab implementation. It would be good if the authors provided some

estimates on how long such strands would need to be in order to have a

reliable numerical interpretation.

It is also not completely clear why a single molecule needs to encode

a number. What happens if multiple strands encoding different numbers

are simultaneously present in solution? If I understood the design

well, this would lead to unpredictable results. So then, it might be

just as well to use a statistical mix of monomers to encode a number.

The paper is exceptionally well written and easy to follow.

Some suggestions for improvement:

- Move the AND gate proof example (Section 5.1) to before the general case

(or at least advertise that the example will come so the reader who likes

examples can read it first)

- The abstract and the introduction do not announce that random bit strings

will be realized as DNA strands, so the reader is left wondering until the

middle of Section 7 about this, that is, where will all these bit streams

be coming from, and it is unclear that anything of the sort exists in biology.

Corrections:

line 125: all the outputs -> all the inputs

line 231: Eq.20 applies for all t>0 -> please rephrase this; Eq 20 contains a

limit so this does not make sense, better to introduce a new equation here

line 272: than -> that

line 284: f -> if

lines 409-410: "or the reactant concentration and can easily be broken up" :

I could not parse this sentence

Reviewer #2: The authors propose a technique for translating a function computed by

stochastic logic circuits into (robust) chemical reaction networks (CRNs).

Further, it is suggested that these CRNs could be implemented using DNA strand

displacements and DNA units called concatemers.

To be more precise, the central result of the paper is a constructive proof of

the fact that any polynomial function, derived from the truth table of a

Boolean function, can be computed by a CRN. Some examples, comprising polynomial

approximations of trascendental functions, are given in the supporting material.

The paper generalizes and simplifies two previous works, cited as Refs.[25] and

[26]. Indeed, both the logic and the biologic techniques are taken from previous

works published in the literature: for example, the synthesis technique for

stochastic logic is based on the framework presented in Refs.[12,13]. Despite

this, I think that the current work is interesting, and deserves to be published.

However, before accepting the paper some aspects of the work should be better

explained or discussed.

First of all, it is not clear how long are the bitstreams used to represent the

(probability of) real-valued variables. This is reflected on the implementation

of CRNs by DNA strands, displacements, and concatemers: how long they are? How

many different complexes are needed?

Related to this, observe that the truth table of a Boolean function of n

Boolean variables contains 2^n rows, and that also the number of minterms may

be exponential in n. When translated to a polynomial, we may easily obtain an

exponential number of terms. This makes the technique infeasible in practice.

How is this problem dealt with?

Maybe this problem could be overcome by decomposing the CRN into a "circuit" of

simpler CRNs, each one computing a simpler function, just like a Boolean circuit

is composed of layers of gates. But then, it is not clear how the output of a

CRN can be fed as input to another CRN. And, sometimes, the output of a CRN must

be duplicated before sending it to the CRNs of the next layer. Albeit the

proposed work is speculative, rather than being a working model, I think that

these aspects should be discussed. It is not clear to me whether the last

paragraph of Section 7, which speaks about multimolecular reactions, is somehow

related with these aspects.

In the Abstract it is stated that the computations performed by the CRNs are

robust to variations in the reaction rates to within a log-order constraint.

However, in Section 6 only a brief discussion stemming from some computer

experiments is presented, and these experiments are only about one specific

polynomial function; there is no proof of the above statement.

Further, it is not clear how these simulations are performed; for example, at

the end of page 7, how can we see that the output value is c = 0.462 instead of

the expected value 0.42?

In equation (16), what kind of operation is x in a x b? Is it just a

multiplication between real numbers?. Rather, I would have said that if we

simulate the CRN we obtain a value c that corresponds to a multiplied by b.

Before Section 6, it is assumed that all reactions have the same reaction rate

k. Having exactly the same reaction rate is an overkill requirement, which is

impossible to obtain in practice, with real CRNs. On the other hand, fortunately,

case 5 on page 16 is not so bad in terms of error; I think that this case should

be taken as a reference for a real biological implementation. However, the fact

that these results have been obtained considering only a specific function, make

me wonder about their validity in general.

At the beginning of Section 6.1, it should be recalled how the error is defined;

this is made only in the caption of Figure 2.

Some typos, and suggestions for local corrections and improvements, follow:

- page 14, line 1: f -> if

- equations (34), (36), and (43): Explicitly say what the three dots at the

beginning of the equations mean

- line 292: We compare -> We compared

- line 337: In Section 7.1 -> This is Section 7.1!

- line 340: Nucleotide -> A nucleotide

- line 355: stages -> stage

- line 375: molecules -> molecule

- page 18, line 4 of caption of Figure 3: reactant -> reactants

- Ref.[8]: still to appear?

- Ref.[12]: venue is missing

- Ref.[46]: some problems in the title

6. PLOS authors have the option to publish the peer review history of their article (what does this mean?). If published, this will include your full peer review and any attached files.

Reviewer #1: No

Reviewer #2: No

---

## [Author Response · Author response to Decision Letter 0]

23 Jan 2023

All information is contained in the "Response to Reviewers" document that we uploaded.

---

## [Editor Report · Decision Letter 1]

27 Jan 2023

Computing Mathematical Functions with Chemical Reactions via Stochastic Logic

PONE-D-22-24554R1

Dear Dr. Riedel,

We’re pleased to inform you that your manuscript has been judged scientifically suitable for publication and will be formally accepted for publication once it meets all outstanding technical requirements.

Kind regards,

Ivan Kryven

Academic Editor

PLOS ONE
---

## [Editor Report · Acceptance letter]

1 Feb 2023

PONE-D-22-24554R1 

Computing Mathematical Functions with Chemical Reactions via Stochastic Logic 

Dear Dr. Riedel:

I'm pleased to inform you that your manuscript has been deemed suitable for publication in PLOS ONE. Congratulations! Your manuscript is now with our production department. 

Kind regards, 

on behalf of

Dr. Ivan Kryven 

Academic Editor

PLOS ONE